# TRiC controls transcription resumption after UV damage by regulating Cockayne syndrome protein A

Alex Pines[1,2], Madelon Dijk[1], Matthew Makowski[3], Elisabeth M. Meulenbroek[4], Mischa G. Vrouwe[1], Yana van der Weegen [1], Marijke Baltissen[3], Pim J. French[5], Martin E. van Royen [6], Martijn S. Luijsterburg[1], Leon H. Mullenders[1], Michiel Vermeulen [3], Wim Vermeulen[2], Navraj S. Pannu[4] & Haico van Attikum[1]

Transcription-blocking DNA lesions are removed by transcription-coupled nucleotide excision repair (TC-NER) to preserve cell viability. TC-NER is triggered by the stalling of RNA polymerase II at DNA lesions, leading to the recruitment of TC-NER-specific factors such as the CSA–DDB1–CUL4A–RBX1 cullin–RING ubiquitin ligase complex (CRL$^{CSA}$). Despite its vital role in TC-NER, little is known about the regulation of the CRL$^{CSA}$ complex during TC-NER. Using conventional and cross-linking immunoprecipitations coupled to mass spectrometry, we uncover a stable interaction between CSA and the TRiC chaperonin. TRiC's binding to CSA ensures its stability and DDB1-dependent assembly into the CRL$^{CSA}$ complex. Consequently, loss of TRiC leads to mislocalization and depletion of CSA, as well as impaired transcription recovery following UV damage, suggesting defects in TC-NER. Furthermore, Cockayne syndrome (CS)-causing mutations in CSA lead to increased TRiC binding and a failure to compose the CRL$^{CSA}$ complex. Thus, we uncover CSA as a TRiC substrate and reveal that TRiC regulates CSA-dependent TC-NER and the development of CS.

[1] Department of Human Genetics, Leiden University Medical Center, Einthovenweg 20, Leiden 2333 ZC, The Netherlands. [2] Department of Molecular Genetics, Cancer Genomics Netherlands, Erasmus University Medical Center, Wytemaweg 80, 3015 CN Rotterdam, The Netherlands. [3] Department of Molecular Biology, Radboud Institute for Molecular Life Sciences, Radboud University Nijmegen, Geert Grooteplein 28 6525 GA Nijmegen The Netherlands. [4] Department of Biophysical Structural Chemistry, Gorlaeus Laboratories, Leiden University, Einsteinweg 55 2333 CC Leiden The Netherlands. [5] Department of Neurology, Cancer Treatment Screening Facility (CTSF), Erasmus University Medical Center, Wytemaweg 80, 3015 CN Rotterdam, The Netherlands. [6] Department of Pathology, Cancer Treatment Screening Facility (CTSF), Erasmus Optical Imaging Centre (OIC), Erasmus University Medical Center, Wytemaweg 80, 3015 CN Rotterdam The Netherlands. These authors contributed equally: Alex Pines, Madelon Dijk. Correspondence and requests for materials should be addressed to W.V. (email: w.vermeulen@erasmusmc.nl) or to N.S.P. (email: raj@chem.leidenuniv.nl) or to H.v.A. (email: h.van.attikum@lumc.nl)

Environmental pollutants, radiation, and cellular metabolites have the propensity to damage DNA and promote genome instability and age-related diseases[1]. The versatile nucleotide excision repair (NER) pathway is an important defense mechanism, which removes a remarkably wide spectrum of DNA-helix destabilizing lesions, including those induced by UV irradiation, via two distinct damage-recognizing sub-pathways: global genome NER (GG-NER) and transcription-coupled NER (TC-NER). While GG-NER removes DNA damage from the entire genome, TC-NER specifically targets transcription-blocking DNA lesions, thereby preserving transcription programs[2,3]. TC-NER is initiated by the stalling of RNA polymerase II at DNA lesions. This triggers the recruitment of the SNF2/SWI2 ATPase CSB and the CSA protein, which promote the assembly of a large repair complex that unwinds the damaged DNA, excises a single-stranded DNA region containing the lesion, and promotes DNA synthesis and ligation to seal the gap[4,5].

CSA comprises a seven-bladed WD40 propeller that, through interactions with DDB1, assembles into a cullin-RING ubiquitin ligase (CRL) complex with CUL4A/B and RBX1 (CRL$^{CSA}$)[6]. CRL$^{CSA}$ binds the COP9 signalosome (CSN) complex[7], which renders CUL4A inactive through deneddylation. Following UV damage, COP9 is likely displaced by CSB when CSA becomes incorporated into the TC-NER complex, triggering CUL4A activation by neddylation[6]. This process is thought to lead to polyubiquitination and subsequent proteasome-dependent degradation of CSB[6,8]. UVSSA on the other hand stabilizes CSB by counteracting its CSA-dependent ubiquitylation by recruiting the broad-spectrum deubiquitinating enzyme USP7[9–11]. In this way, CRL$^{CSA}$ and UVSSA-USP7 act antagonistically to coordinate the timely removal of CSB from transcription-blocking lesions, allowing efficient restart of transcription following TC-NER.

Genetic defects in *CSA* and *CSB* mostly give rise to Cockayne syndrome, which is a multisystem-disorder characterized by premature aging, progressive mental and sensorial retardation, microcephaly, severe growth failure, and cutaneous photosensitivity[12]. Despite the important role of CSA in controlling TC-NER and preventing adverse effects on health, remarkably little is known about the regulation of CSA in the context of the CRL$^{CSA}$ complex.

Here we use conventional and cross-linking immunoprecipitations coupled to mass spectrometry to uncover proteins that bind and regulate the function of CSA. Using this approach, we identify several new CSA-interacting proteins, including all subunits of the TRiC complex. TRiC is a eukaryotic chaperonin that has evolved to ensure proteome integrity of essential and topologically complex proteins, including cell-cycle regulators, signaling proteins, and cytoskeletal components[13,14]. We find that TRiC's binding to CSA ensures its proper folding and DDB1-dependent assembly into the CRL$^{CSA}$ complex. Consequently, loss of functional TRiC affects CSA's localization and stability, and impairs transcription recovery after DNA damage induction. These findings show that CSA is a TRiC substrate and reveal a role for the TRiC chaperonin in regulating CSA-dependent TC-NER.

## Results

**CSA interacts with chaperonin TRiC.** To identify CSA regulating proteins, we stably expressed FLAG-tagged CSA in CSA-deficient patient cells (CS3BE-SV40), and performed a pulldown of CSA-FLAG followed by mass spectrometry (MS). Among the top hits were known interactors of CSA, such as the members of the COP9 signalosome (e.g., COPS2 and COPS3) and the CRL$^{CSA}$ complex (e.g., DDB1 and CUL4A), as well as the TC-

NER proteins CSB and UVSSA[2,6,7,15] (Supplementary Data 1). Unexpectedly, our approach also identified all eight subunits of the TRiC chaperonin complex as CSA-interacting factors (Fig. 1a and Supplementary Data 1). A FLAG pulldown from cells expressing CSA-FLAG followed by western blot analysis confirmed the interaction between CSA and the TRiC subunit TCP1 (Fig. 1b). Moreover, immunoprecipitation of CSA from human fibroblasts followed by western blot analysis confirmed a UV-independent interaction between CSA and TCP1 at the endogenous level, as well as the known UV-dependent interaction with the elongating form of RNAPII (RNAPIIo)[16] (Fig. 1c). Finally, pulldown of CSA-GFP from CSA-deficient patient cells confirmed interactions between CSA and the TRiC subunits CCT4 and CCT5 (Fig. 1d, e). These results demonstrate that CSA interacts with the TRiC complex.

We then addressed if the CSA-TRiC complex is distinct from the CRL$^{CSA}$ complex by performing a tandem pulldown of CSA-FLAG and DDB1-GFP from U2OS cells that co-expressed these fusion proteins. Pulldown of CSA-FLAG confirmed interactions with both GFP-DDB1 and CUL4A, as well as TRiC components CCT4 and CCT7 (Fig. 1e). Importantly, subsequent specific enrichment of CRL$^{CSA}$ by pulldown of GFP-DDB1 revealed an interaction with CUL4, but not with CCT4 or CCT7 (Fig. 1e). We therefore conclude that TRiC preferentially interacts with CRL-free CSA.

**CSA binds the inner pocket of TRiC.** TRiC/CCT (TCP1 ring complex/chaperonin containing TCP1) is an ATP-dependent complex composed of two stacked octameric rings. Each ring consists of eight different but related subunits, which are present once per ring[17]. Moreover, each ring creates an inner pocket where substrate proteins interact to become properly folded[18,19]. To gain more insight into the interaction between CSA and TRiC, we stably expressed CSA-GFP in CSA-deficient patient cells, and identified CSA interacting proteins using a label-free quantification (LFQ), GFP-Trap affinity purification (AP)-MS/MS approach (Fig. 2a). Even after stringent washing at 1 M NaCl and 1% NP-40, the interaction between CSA and DDB1, CUL4A, RBX1, and members of the COP9 signalosome was preserved. Importantly, the LFQ analysis also detected all subunits of the TRiC complex, indicating that the CSA–TRiC interaction is highly stable. Moreover, the use of ethidium bromide excludes the possibility that these interactions are mediated by DNA, which is in agreement with our observation that most CSA-TRiC complexes are found in the soluble fraction of the cell (Fig. 1b, c). Finally, we used an iBAQ-based method[20] to estimate the relative stoichiometries of the various proteins immunoprecipitated by CSA. This revealed an interaction stoichiometry of ~1 TRiC subunit per 3 CSA proteins (Fig. 2b).

To examine whether the strong nature of the CSA–TRiC interaction is mediated by other proteins or can be ascribed to direct binding of CSA to TRiC, we applied xIP-MS[21]. Immunoprecipitation of CSA-GFP by GFP-TRAP was followed by on-bead cross-linking and tryptic digestion of the bound proteins into covalently cross-linked peptides. Identification of cross-linked peptides was performed using pLink[22] after analysis by mass spectrometry, which revealed residues in close spatial proximity. We identified 149 unique, high confidence residue cross-links in total (Fig. 2c and Supplementary Data 2). Of these, 62 linkages were intra- or inter-linkages mapping to subunits of the TRiC complex (Supplementary Fig. 1a). All of these TRiC cross-links were consistent with a cross-linker spacer length of less than 34 Å, confirming the structural validity of our data (Supplementary Fig. 1b). Importantly, we observed 11 cross-links between CSA and TRiC subunits CCT3, CCT4, and CCT6

involving CSA residues Lys34, Lys85, Lys167, and Lys212 (Fig. 2c). Although this does not provide information about specific residues that mediate the interaction, the location of these lysine residues in the outer regions of the β-propeller blades made up by the WD40 domain of CSA suggests that these regions are important for the interaction with TRiC (Supplementary Fig. 4a). Given these inter-protein linkages as distance restraints, we used DisVis[23] to identify the accessible interaction space for CSA on the TRiC surface (Fig. 2d). Our data indicate that the only available interaction space for CSA that is consistent with our cross-linking data is within TRiC's inner pocket.

**Loss of TRiC components reduces CSA stability.** TRiC has been described to be involved in the folding or stabilization of ~10% of all newly synthesized proteins[24]. Among the known TRiC substrates are many WD40 repeat-containing proteins. Given that CSA contains seven of such repeats and considering our observation that TRiC directly interacts with CSA, we hypothesized that TRiC could be important for proper folding of CSA and consequently for its stability. To assess this, we depleted TCP1 using siRNAs and examined CSA levels in whole cell extracts by western blot analysis at different times after siRNA transfection (Fig. 3a, b and Supplementary Fig. 2a,b). TCP1 knockdown resulted in a marked decrease in the overall amount of CSA when

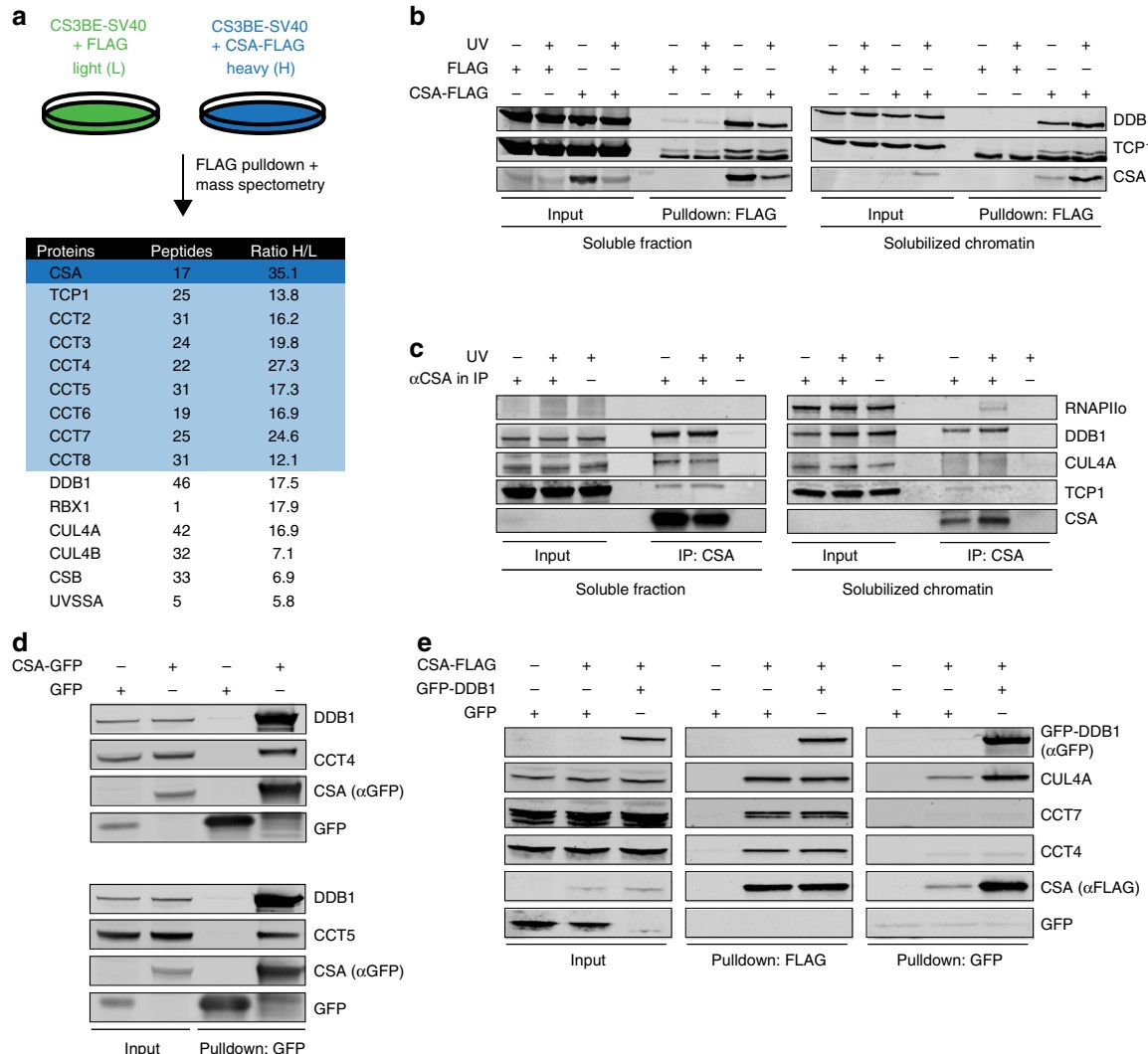

**Fig. 1** CSA interacts with chaperonin TRiC. **a** A SILAC-mass spectrometry approach identified all TRiC subunits as CSA-interacting proteins. CSA-deficient CS3BE-SV40 cells expressing FLAG or CSA-FLAG were cultured in medium containing light or heavy lysine and arginine isotopes, respectively. FLAG- and CSA-FLAG-interacting proteins were pulled down and samples were processed and analyzed by mass spectrometry. The table shows the number of unique peptides found for the top ranked interactors, as well as the ratio of the interactor in the CSA-FLAG pulldown to that in the control FLAG pulldown (ratio H/L). **b** FLAG pulldowns confirm the UV-independent interaction between CSA-FLAG and TCP1. CS3BE-SV40 cells expressing FLAG or CSA-FLAG were mock-treated or UV-C irradiated (20 J/m$^2$). After 1 h of recovery cells were lysed and fractionated into soluble or solubilized chromatin. FLAG pulldowns using both fractions were followed by western blot analysis for the indicated proteins. **c** CSA co-immunoprecipitation confirms the interaction between endogenous CSA and TCP1. As in **b**, except that VH10-hTert cells were used and that endogenous CSA was immunoprecipitated. **d** GFP pulldowns confirm the interaction between CSA and TRiC subunits CCT4 and CCT5. GFP or CSA-GFP was pulled down from CS3BE-SV40 cells. **e** Tandem FLAG and GFP pulldowns show preferential binding of TRiC to DDB1/CUL4A/RBX1-free CSA. CSA-FLAG, GFP, and GFP-DDB1 were expressed in U2OS cells as indicated. Enrichment of CSA-interacting proteins by means of FLAG pulldowns confirmed interactions between CSA and DDB1 and CUL4A, as well as the TRiC subunits CCT4 and CCT7. Subsequently, eluted protein complexes were subjected to pulldown of GFP-DDB1, revealing an interaction with CUL4A, but not CCT4 and CCT7. Full-size scans of western blots are provided in Supplementary Fig. 7

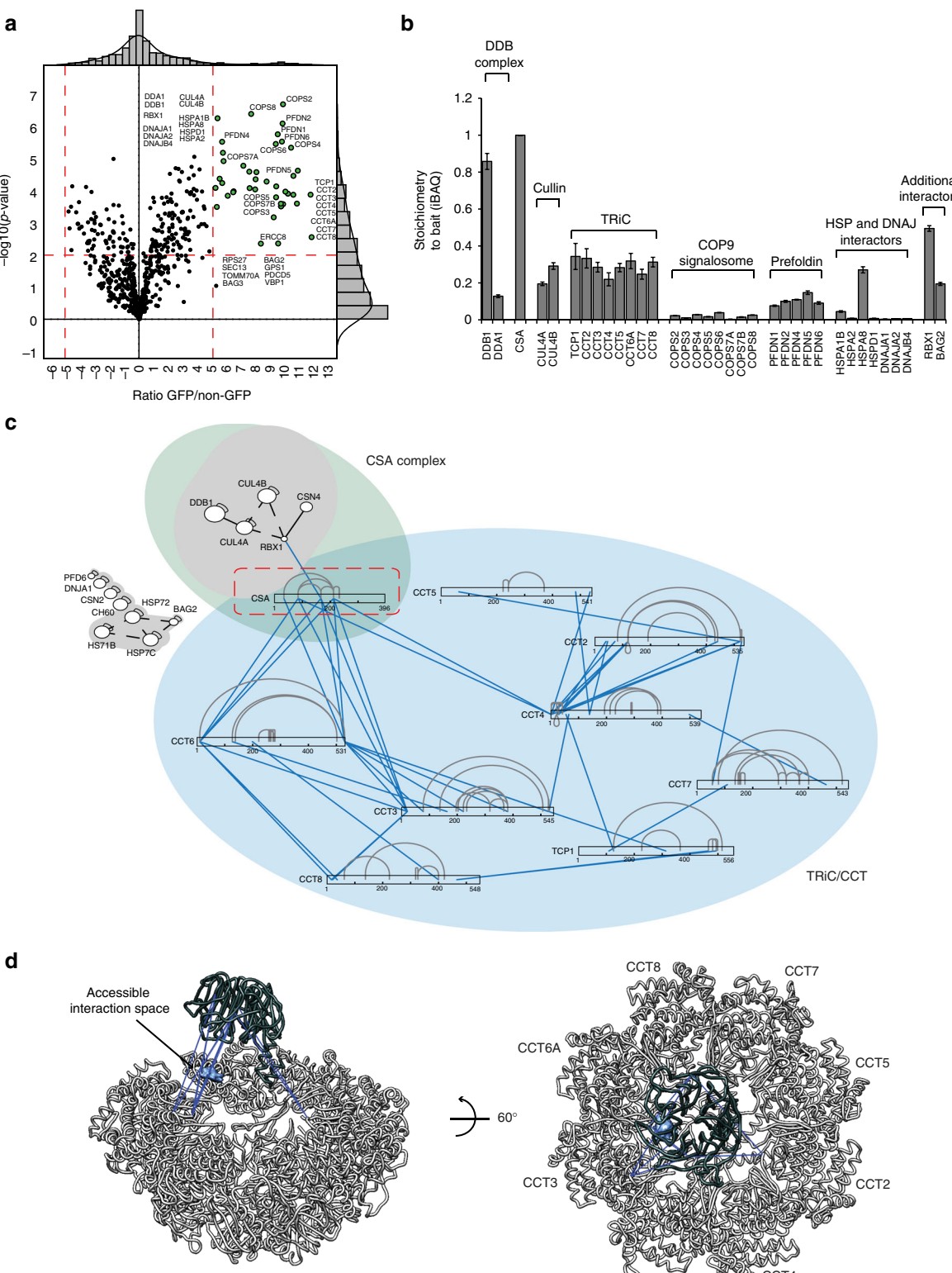

**Fig. 2** xIP-MS reveals that CSA interacts with the TRiC inner pocket. **a** LFQ analysis after CSA-GFP pulldown indicates that all TRiC subunits interact with CSA even after stringent washing. Ratio of protein signal in GFP vs. non-GFP pulldowns is plotted on the *x*-axis, and the significance of the difference, −log₁₀(*p*-value), is plotted on the *y*-axis. Cutoffs are selected such that no protein significantly interacted with the non-GFP control beads. **b** iBAQ-based stoichiometry of selected interactors relative to the bait protein (CSA), which was set to 1. **c** Cross-linking map of all identified residue linkages. TRiC subunits are displayed in linear form with intra-links indicated in gray. The presence of ambiguous linkages (where multiple subunits have the same peptide) is indicated by dashed lines. Inter-protein linkages are indicated in blue. **d** CSA inter-protein linkages with the TRiC octamer (colored light gray) indicate that CSA binds the TRiC inner pocket. Inter-protein cross-links are colored dark-blue. CSA (colored dark gray) was positioned manually to give a visual interpretation to possible CSA-TRiC interactions. The accessible CSA interaction space satisfying 10/11 inter-protein cross-links is shown as a light blue cloud

compared to control cells treated with siRNAs against Luciferase, whereas the levels of DDB1 remained unaffected. The reduction in CSA levels correlated with the knockdown efficiency of TCP1.

Knockdown of a single TRiC component has been shown to negatively impact the stability of other subunits in the complex[25],

thereby lowering the availability of functional TRiC complexes in the cell. To confirm that our observations are not specific for TCP1 knockdown, but are the consequence of the loss of TRiC complexes, we also examined the effect of CCT4, CCT5, and CCT7 depletion on CSA protein abundance. Knockdown of these

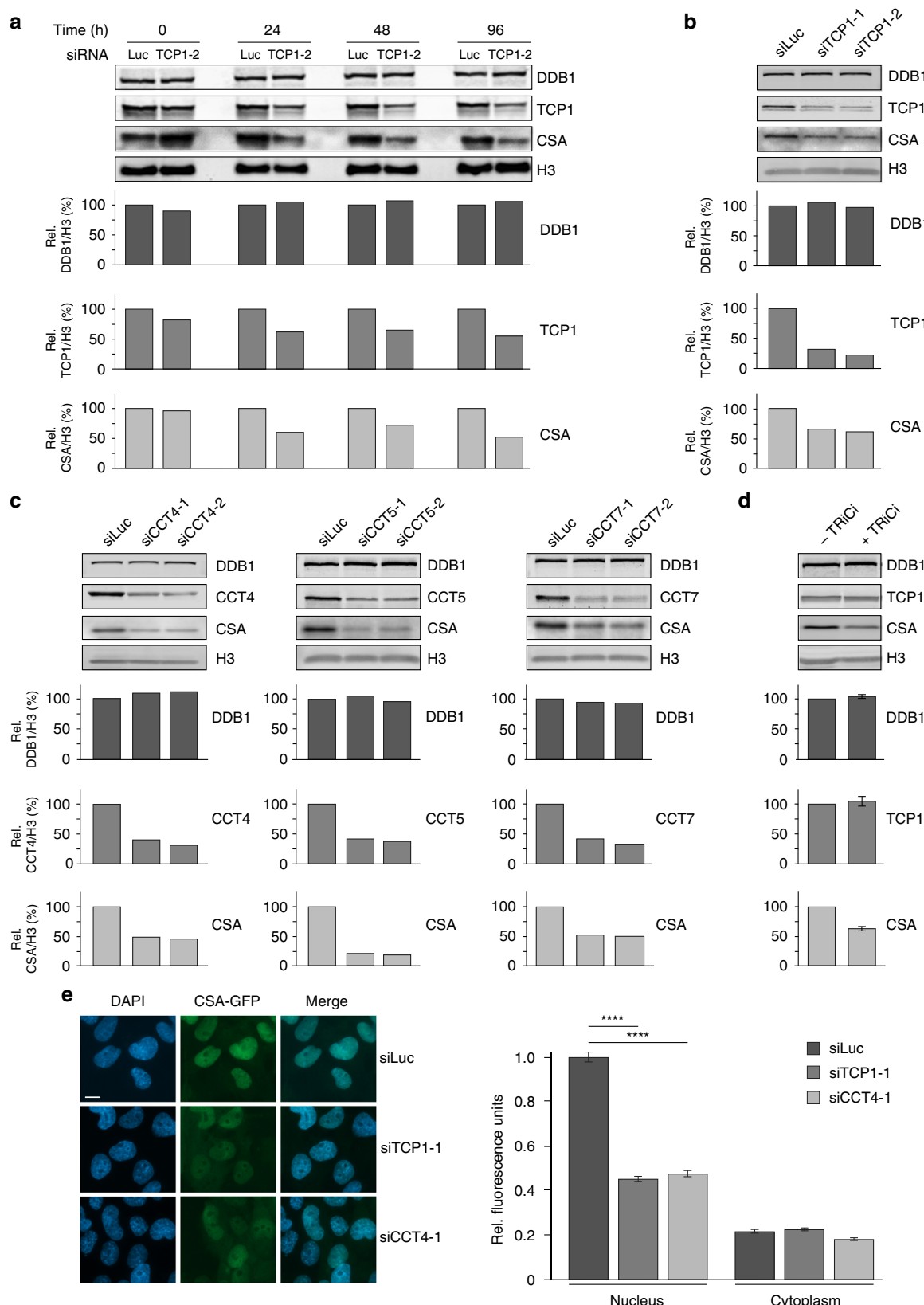

TRiC subunits using different siRNAs also caused a reduction in the CSA levels (Fig. 3c and Supplementary Fig. 2c). Similarly, treatment with a TRiC inhibitor (TRiCi), which has been shown to inhibit archaeal TCP1 activity in vitro[26], led to a substantial decrease in CSA levels while not affecting TCP1 levels itself (Fig. 3d). This shows that CSA stability is not only negatively affected by the loss of TRiC protein, but also by inhibition of its chaperonin activity. To validate these findings, we expressed CSA-GFP in CSA-deficient patient cells and examined the effect of TCP1 and CCT4 knockdown on CSA-GFP expression by fluorescence microscopy analysis. Similar to endogenous CSA, we found that CSA-GFP is primarily expressed in the nucleus. Depletion of either TCP1 or CCT4 significantly reduced the levels of CSA-GFP in the nucleus (Fig. 3e). This reduction in CSA-GFP protein levels is consistent with the effect on endogenous CSA as observed by western blot analysis (Fig. 3a–c and Supplementary Fig. 2). Taken together, these findings indicate that the TRiC complex is involved in regulating CSA stability, likely by affecting proper folding of CSA.

**TRiC is involved in the formation of the CRL^CSA complex.** CSA is a stable component of the DDB1- and RBX1-containing CRL^CSA complex. In this complex, it directly associates with DDB1[6] and likely functions as the substrate receptor. Considering that TRiC is required for CSA stability, we wondered whether DDB1 acts as an acceptor of TRiC-bound CSA in the CRL^CSA complex. To test this, we first pulled down CSA-GFP from CSA-deficient patient cells that were treated with siRNAs against DDB1. Knockdown of DDB1 not only led to a decrease in the association of CSA with DDB1 and CUL4A, but also negatively affected the binding to CSB (Fig. 4a). Strikingly, however, the efficiency by which CSA binds to the TRiC subunit TCP1 appeared to be substantially increased, suggesting that DDB1 may serve as an acceptor of CSA. Secondly, we created a mutant, CSA ΔN, which lacks the first 21 amino acids required for DDB1 binding[6] (Fig. 4b), which was stably expressed in CRISPR/Cas9-mediated CSA knockout U2OS cells (Fig. 4c). Pulldown of GFP-tagged CSA ΔN from these cells not only showed the expected decrease in DDB1 binding as compared to CSA WT, but also abolished the interaction with CSB (Fig. 4d). Importantly, the interaction between CSA ΔN and TCP1 was substantially increased as compared to full-length CSA (Fig. 4d). These results show that interfering with the CSA–DDB1 interaction, either by depletion of DDB1 or deletion of the DDB1-interacting domain in CSA, strongly enhances the interaction between CSA and TRiC. This suggests that in the absence of DDB1, CSA remains tightly bound to the TRiC complex and that DDB1 serves as an acceptor of TRiC-bound CSA in the CRL^CSA complex.

Next, we studied the effect of DDB1 loss on the expression and localization of CSA-GFP following its expression in CSA-deficient patient cells by fluorescence microscopy analysis. DDB1 knockdown led to a significant decrease in nuclear CSA-GFP levels, while CSA-GFP levels in the cytoplasm increased (Fig. 4e), likely due to persistent binding of CSA-GFP to TRiC (Fig. 4a). The latter is consistent with the fact that TRiC is a chaperonin that primarily localizes to and functions in the cytoplasm. Together our findings suggest a hand-over mechanism in which cytoplasmic TRiC provides properly folded CSA to DDB1, thereby facilitating its assembly into CRL^CSA complexes that translocate into the nucleus. Hand-over of CSA might occur directly after its release by TRiC in the cytoplasm, as we detected TRiC-bound, as well as DDB1-bound cytoplasmic CSA (Supplementary Fig. 3).

**A CSA mutant of the top platform shows increased TRiC binding.** The four residues in CSA that were revealed by xIP-MS to be in proximity of the CSA-TRiC binding interface surround a platform at the top of CSA that is formed by the β-propeller blades[6] (Supplementary Fig. 4a and Supplementary Data 2). In order to further assess the functional relevance of the CSA-TRiC interaction, we created eight different CSA mutants in which one of the residues Glu103, Phe120, Lys122, Arg164, Lys247, Lys292, Lys293, or Arg354 in this platform was substituted by Alanine (Supplementary Fig. 4a). Immunoprecipitation of these mutants from CSA-deficient patient cells did not reveal any major difference in their interaction with TCP1, as well as the CRL^CSA complex members DDB1 and CUL4A, as compared to wildtype CSA (Supplementary Fig. 4b). Accordingly, expression of each mutant could also rescue the UV sensitivity of the CSA-deficient patient cells (Supplementary Fig. 4c). Aiming to induce a greater effect on CSA, we next generated a CSA mutant (CSA 8M) that contains all the eight afore-studied mutations in the top platform. Since according to the 3D structure of CSA–DDB1[6] this platform of CSA is not directly involved in DDB1 binding (Figs. 4b and 5a), we expected that the combined eight mutations would leave the CRL^CSA intact (Fig. 5a, b). Surprisingly, however, pulldown of GFP-tagged CSA WT and CSA 8M from CSA-deficient patient cells showed decreased binding of CSA 8M to CSB, DDB1, and CUL4A when compared to CSA WT (Fig. 5c). This indicated that the mutations impacted CSA's interactions in a manner similar to DDB1 depletion or deletion of the DDB1-interacting domain in CSA (Fig. 4a, d). We therefore wondered whether the altered interactions observed for CSA 8M could be explained by, or lead to a change in TRiC binding. Indeed, CSA 8M showed greatly increased binding to TCP1 when compared to CSA WT (Fig. 5c). Given that the mutated residues do not directly bind to DDB1, we consider it most plausible that the mutations negatively affect the release of CSA by TRiC. This is strengthened by fluorescence microscopy-based analysis of CSA 8M expression, which revealed that this mutant largely fails to localize to the nucleus and remains mainly cytoplasmic (Fig. 5d), a phenotype reminiscent of that observed after DDB1 knockdown (Fig. 4e). This corroborates

**Fig. 3** Loss of TRiC components reduces CSA stability. **a** Depletion of TCP1 decreases CSA protein abundance. VH10-hTert cells were transfected with the indicated siRNAs and total cell extracts were prepared at the indicated time points after siRNA transfection. Protein levels were determined by western blot analysis of the indicated proteins. H3 is a loading control. Graphs represent the ratio of protein signal intensities over H3 control signal intensities for siTCP1-treated cells relative to that for siLuc-treated control cells, which was set to 100%, at each time point. A repeat of the experiment is shown in Supplementary Figure 2a. **b** Depletion of TCP1 decreases CSA protein abundance. As in **a**, except that two different siRNAs against TCP1 were used and that protein levels were determined 72 h after siRNA transfection. A repeat of the experiment is shown in Supplementary Figure 2b. **c** Depletion of CCT4, CCT5, or CCT7 decreases CSA protein abundance. As in **a**, except that CCT4, CCT5, or CCT7 siRNAs were used and that protein levels were determined 72 h after siRNA transfection. A repeat of the experiment is shown in Supplementary Figure 2c. **d** TRiC inhibition decreases CSA protein abundance. VH10-hTert cells were treated with DMSO or an inhibitor against the TRiC subunit TCP1 (TRiCi). Protein levels were determined after 72 h of treatment. **e** TCP1 or CCT4 loss decreases CSA-GFP protein abundance in the nucleus. TCP1 or CCT4 was depleted from CSA-GFP expressing CS3BE-SV40 cells using the indicated siRNAs. Nuclear and cytoplasmic CSA-GFP levels were analyzed and quantified by fluorescence microscopy and ImageJ. GFP signal intensities were normalized to the average nuclear signal in siLuc-treated cells. Data represent mean ± SEM of 190 cells quantified in two independent experiments. p-Values were derived from an unpaired t-test. Length of scale bar: 10 μm. Full-size scans of western blots are provided in Supplementary Fig. 8

our conclusion that cytoplasmic TRiC provides properly folded CSA to DDB1 for incorporation into CRL$^{CSA}$ complexes and subsequent translocation into the nucleus.

**Loss of TRiC reduces RRS and protection against UV damage.** The CRL$^{CSA}$ complex is a nuclear core component of the TC-

NER machinery. Since TRiC is critical for regulating CSA stability and formation of the CRL$^{CSA}$ complex, we asked if the TRiC-dependent regulation of CSA is a prerequisite for functional TC-NER. Indeed, we found that the recovery of RNA synthesis (RRS) after global UV-irradiation, which is an established measure for TC-NER, was impaired in TCP1-depleted cells when compared to control cells (Fig. 6a), while basal transcription levels remained

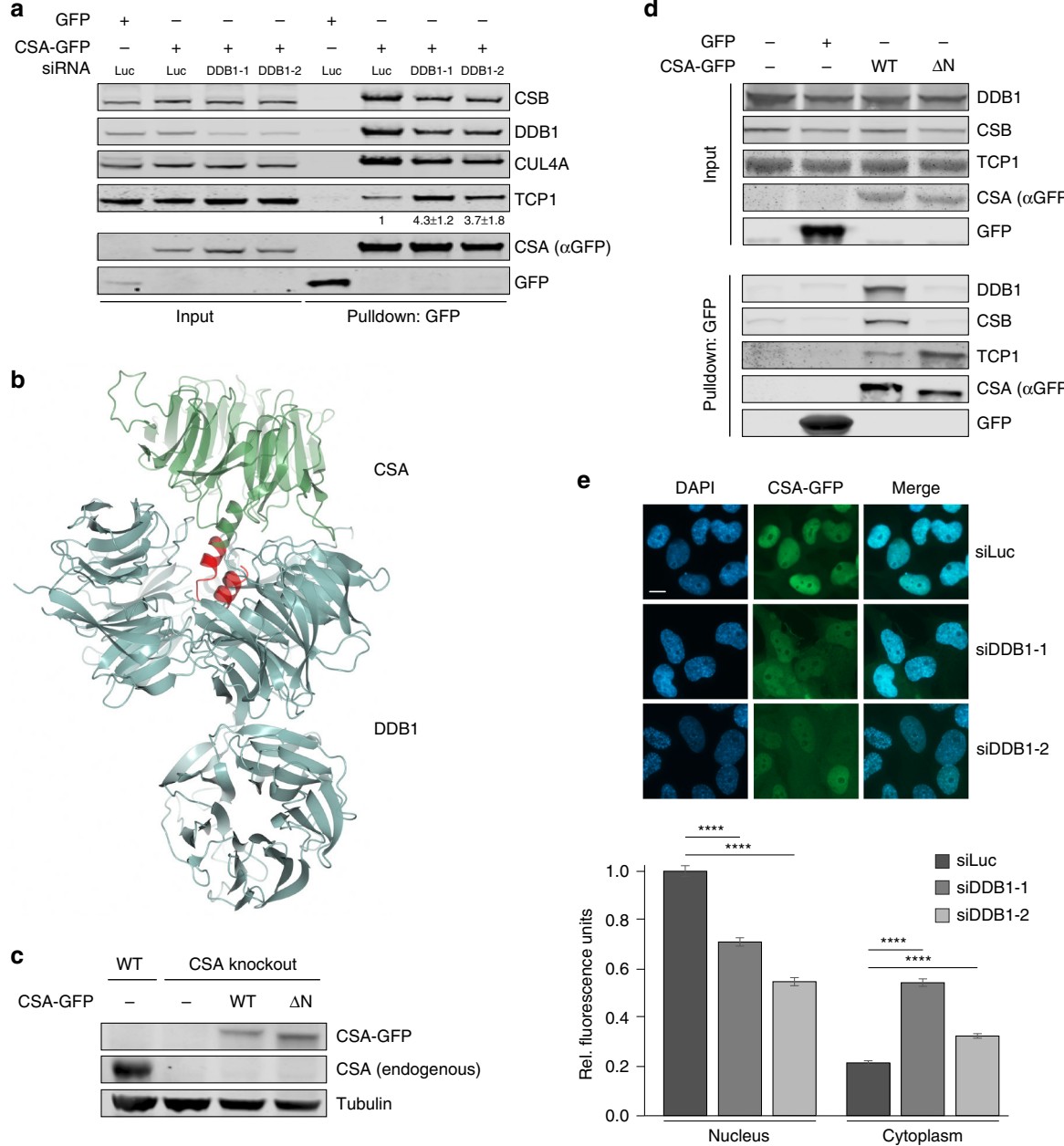

**Fig. 4** TRiC is involved in the formation of the CRL$^{CSA}$ complex. **a** DDB1 loss enhances the interaction between TCP1 and CSA. CSA-GFP was pulled down from CS3BE-SV40 cells treated with the indicated siRNAs. Protein levels were determined by western blot analysis of the indicated proteins. The ratio of TCP1 signal intensities over CSA for siDDB1-treated cells relative to that for siLuc-treated control cells, which was set to 1, is shown as the mean ± SEM of three independent experiments. **b** Overall structure of CSA (green) bound to DDB1 (blue), showing that CSA's N-terminus is directly involved in DDB1 binding. CSA ΔN lacks amino acids 1–21, which are shown in red. Visualization was done in ccp4mg using structure 4a11 from the PDB. Length of scale bar: 10 μm. **c** Stable expression of CSA-GFP WT or CSA-GFP ΔN in CSA knockout U2OS. Protein levels were determined by western blot analysis of the indicated proteins. Tubulin is a loading control. **d** Deletion of CSA's DDB1-interacting domain leads to increased TRiC binding. Stably expressed GFP-NLS, CSA-GFP WT, and CSA-GFP ΔN were pulled down from CSA knockout U2OS cells as indicated. **e** DDB1 decreases CSA-GFP protein abundance in the nucleus concomitantly with an increase in cytoplasmic localization. DDB1 was depleted from CSA-GFP expressing CS3BE-SV40 cells using the indicated siRNAs. Nuclear and cytoplasmic CSA-GFP levels were analyzed and quantified by fluorescence microscopy and ImageJ. GFP signal intensities were normalized to the average nuclear signal in siLuc-treated cells. Data represent mean ± SEM of 190 cells quantified in two independent experiments. p-Values were derived from an unpaired t-test. Full-size scans of western blots are provided in Supplementary Fig. 9

unaffected by TCP1 knockdown (Supplementary Fig. 5a). A similar effect on RRS could be observed after knockdown of CCT4, CCT5, or CCT7 (Supplementary Fig. 5b). In contrast, depletion of several individual TRiC subunits did not affect GG-NER, as determined by measuring DNA repair synthesis (Supplementary Fig. 5c,d). Furthermore, we found that in CSA-deficient patient cells expressing CSA 8M RRS was reduced when compared to that in cells expressing CSA WT (Fig. 6b), showing that not only CSA instability, but also persistent binding of CSA to TRiC negatively impacts TC-NER. In agreement with a defect in TC-NER, we also observed that TCP1-depleted cells, as well as cells depleted of several other individual TRiC subunits, were markedly more sensitive to UV when compared to control cells as measured in alamarBlue-based viability assays (Fig. 6c and Supplementary Fig. 6a). Notably, overexpression of CSA partially

alleviated the UV sensitivity of TCP1-depleted cells, suggesting that this phenotype is largely due to loss of CSA stability and not that of another TRiC substrate (Supplementary Fig. 6b). Moreover, expression of mutant CSA 8M in patient cells failed to complement the relatively high UV sensitivity caused by CSA deficiency, whereas expression of CSA WT could do so, as determined in clonogenic survival assays (Fig. 6d, Supplementary Fig. 6c). Finally, expression of CSA ΔN in CSA knockout U2OS cells could not rescue the extreme sensitivity of these cells to Illudin S, which is an agent that induces transcription-blocking DNA lesions that are repaired by TC-NER[27], whereas expression of CSA WT fully rescued this phenotype (Fig. 6e). Together these data show that TRiC, by regulating CSA stability and incorporation into the CRL$^{CSA}$ complex, promotes TC-NER and protects cells against UV-induced damage.

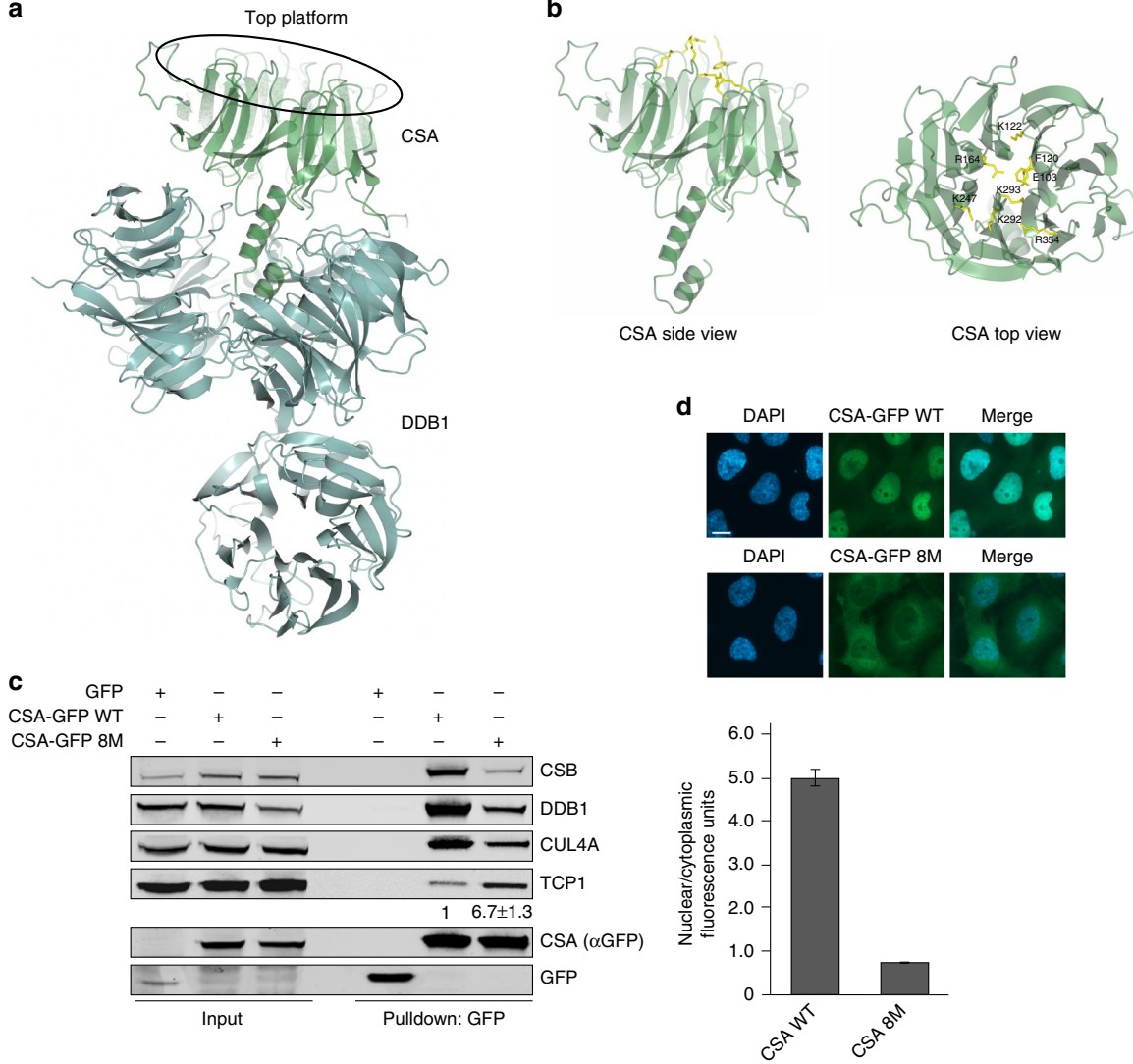

**Fig. 5** A CSA mutant of the top platform shows increased TRiC binding. **a** Overall structure of CSA (green) bound to DDB1 (blue), showing that not CSA's top platform, but its N-terminus is directly involved in DDB1 binding. Visualization was done in ccp4mg using structure 4a11 from the PDB. **b** Side and top view of CSA. The amino acids Glu103, Phe120, Lys122, Arg164, Lys247, Lys292, Lys293, and Arg354 in CSA's top platform that were mutated to Alanines in the CSA 8M mutant are shown in yellow. **c** The CSA 8M mutant shows decreased incorporation into the CRL$^{CSA}$ complex, but increased TCP1 binding. CSA-GFP WT and CSA-GFP 8M were pulled down from CS3BE-SV40 cells. Protein levels were determined by western blot analysis of the indicated proteins. The ratio of TCP1 signal intensity over CSA-GFP 8M relative to that of TCP1 over CSA-GFP WT, which was set to 1, is shown as the mean ± SEM of two independent experiments. **d** CSA-GFP 8M shows reduced protein abundance in the nucleus concomitantly with an increase in cytoplasmic localization. Mean nuclear and cytoplasmic GFP levels were analyzed and quantified by fluorescence microscopy and ImageJ. For each cell the nuclear/cytoplasmic ratio was calculated. Data represent mean ± SEM of 160 cells quantified in two independent experiments. Length of scale bar: 10 μm. Full-size scans of western blots are provided in Supplementary Fig. 10

**Patient mutations in CSA cause increased TRiC binding.** Mutations in the *CSA* gene have been found to underlie the multi-system disorder Cockayne syndrome (CS). CS patients suffer from cutaneous photosensitivity and severe neurological and developmental defects[12]. Although part of the cases can be explained by mutations that lead to a non-functional and/or truncated CSA protein, it remains to be established how a group of single missense mutations can give rise to CS. Importantly, the majority of these mutations are present in the WD40 repeats of CSA that we discovered to be important for the interaction with TRiC (Fig. 2 and Supplementary Fig. 4a). To unravel the effect of such disease-causing point mutations on the CSA protein, we created GFP-tagged CSA constructs harboring patient mutations A160T, A205P, or D266G, which are found in WD40 repeats 3, 4, and 5, respectively (Fig. 7a)[28]. A160T and A205P have been predicted to interfere with the integrity of the overall fold,

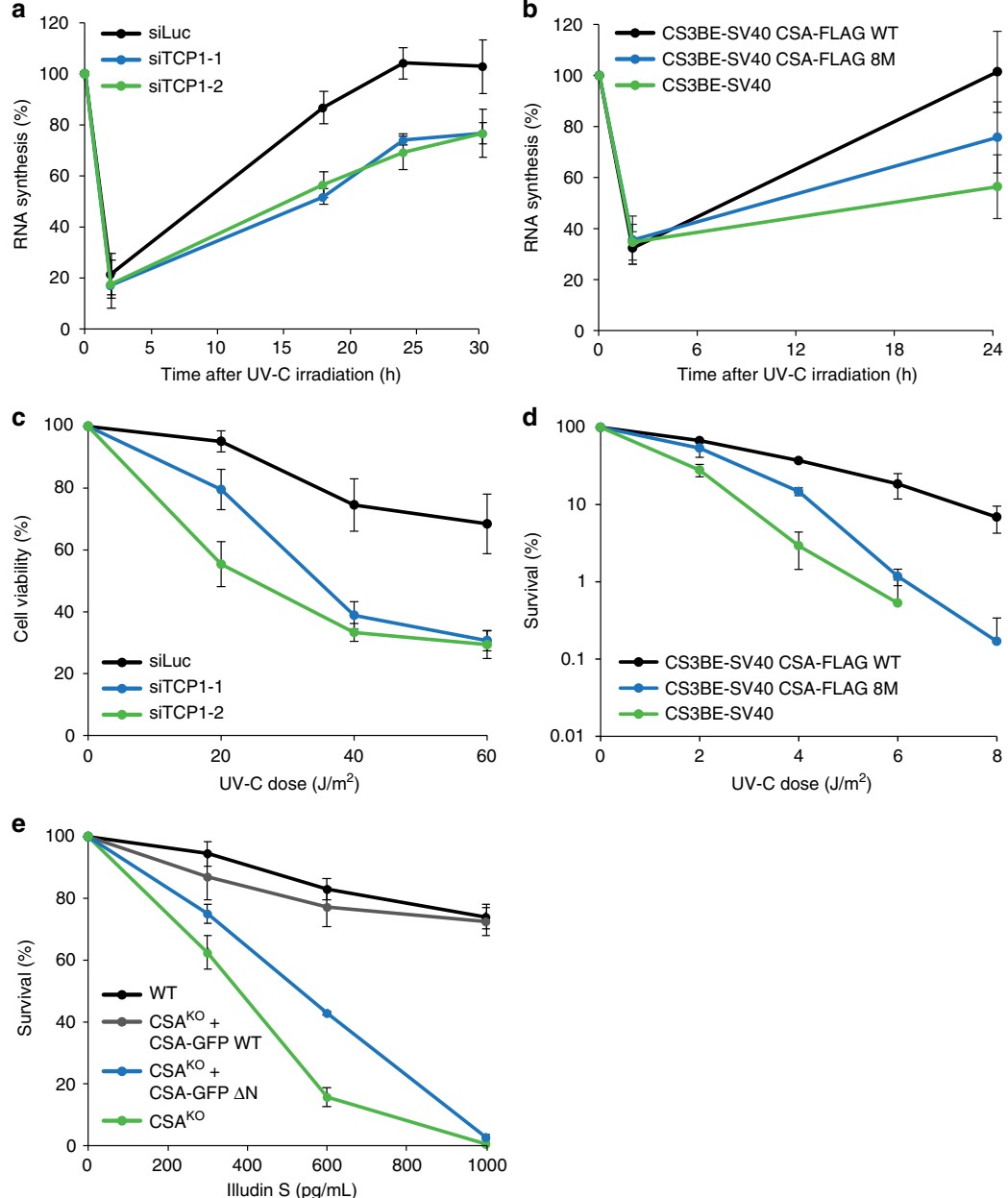

**Fig. 6** Loss of TRiC reduces RRS and protection against UV damage. **a** TCP1 loss reduces RNA synthesis recovery following UV-C irradiation. VH10-hTert cells were transfected with the indicated siRNAs and UV-C irradiated (10 J/m²). RNA synthesis was measured by means of EU incorporation at the indicated time points after UV. RNA synthesis levels were normalized to those in non-irradiated cells, which were set to 100%. Data represent the mean ± SEM of four independent experiments. **b** Expression of CSA-FLAG 8M shows reduced RNA synthesis recovery as compared to expression of CSA-FLAG WT. As in **a**, except that CS3BE-SV40 cells expressing CSA-FLAG WT or CSA-FLAG 8M were used. Data represent the mean ± SEM of four independent experiments. **c** TCP1 loss renders cells hypersensitive to UV damage. VH10-hTert cells were transfected with the indicated siRNAs, UV-C irradiated at the indicated doses and 72 h later assayed for viability using alamarBlue®. Data represent mean ± SEM of four independent experiments. **d** Expression of CSA-FLAG 8M in CS3BE-SV40 cells fails to rescue UV-sensitivity. CS3BE-SV40 cells stably expressing CSA-FLAG WT or CSA-FLAG 8M were UV-C irradiated and clonogenic survival was measured. Data represent mean ± SEM of three independent experiments. **e** CSA WT, but not CSA ΔN, complements the Illudin S sensitivity of CSA knockout (KO) U2OS cells. The indicated cells were treated with different concentrations of Illudin S and clonogenic survival was determined. Data represent mean ± SEM of three independent experiments

whereas D266G is expected to have mostly local effects[6]. Interestingly, pulldown of these mutants from U2OS cells revealed substantially increased TRiC binding as compared to wild-type CSA, suggesting misfolding of the mutated CSA proteins (Fig. 7b). Moreover, none of the three mutants appeared to adopt a conformation suitable for incorporation into the CRL$^{CSA}$ complex, as reflected by the lack of DDB1 and CUL4A binding. Fluorescence microscopy further illustrated that whereas wildtype CSA was translocated into the nucleus, all three mutants were predominantly present in the cytoplasm (Fig. 7c), indicating that these patient mutations lead to a CSA protein that fails to localize to the nucleus. Thus, we provide evidence that disease-associated missense mutations in CSA can lead to enhanced interaction with TRiC and cause cellular mislocalization. This underscores the importance of the TRiC chaperonin in CSA folding/stabilization and assembly of the CRL$^{CSA}$ complex, as well as in the development of CS.

## Discussion

A network of chaperones and protein degradation machineries, called the proteostasis network (PN) is required to maintain

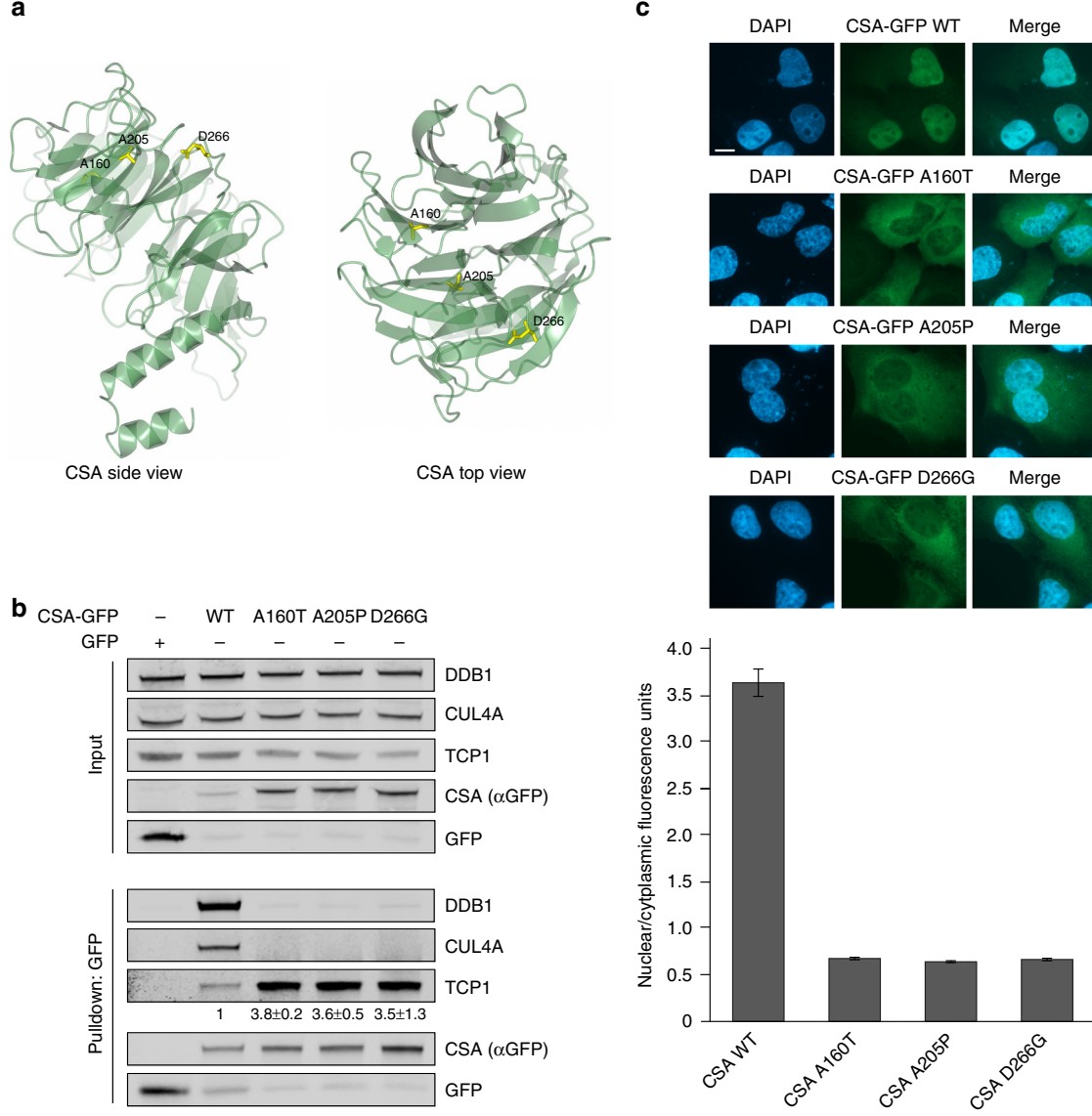

**Fig. 7** Patient mutations in CSA cause increased TRiC binding. **a** Side and top view of CSA. Residues Ala160, Ala205, and Asp266 that have been found mutated in Cockayne syndrome patients are shown in yellow. Visualization was done in ccp4mg using structure 4a11 from the PDB. **b** CSA harboring patient mutation A160T, A205P, or D266G shows increased binding to TRiC and failure to be incorporated into the CRL$^{CSA}$ complex. CSA-GFP WT and CSA-GFP containing the indicated mutations were pulled down from U2OS cells. Protein levels were determined by western blot analysis. The signal intensity ratio of TCP1 over the CSA-GFP mutant relative to that of TCP1 over CSA-GFP WT, which was set to 1, is shown as the mean ± SEM of two independent experiments. **c** CSA A160T, A205P, and D266G show predominant cytoplasmic localization. CSA-GFP WT and CSA-GFP containing the indicated mutations were expressed in U2OS. Mean nuclear and cytoplasmic GFP intensities were analyzed and quantified by fluorescence microscopy and ImageJ. For each cell, the nuclear/cytoplasmic ratio was calculated. Data represent mean ± SEM of 100 cells quantified in two independent experiments. Length of scale bar: 10 μm. Full-size scans of western blots are provided in Supplementary Fig. 10

protein homeostasis[29]. By regulating protein stability and degradation in cells, the PN drives vital processes[30]. Although several components of the PN have been found implicated in the DNA damage response[31–34], mechanistic insight into how this network affects these processes has remained largely elusive. Here we demonstrate that one of the components of the PN, the chaperonin TRiC, stably interacts with the core TC-NER protein CSA. By encapsulating CSA in its inner pocket, TRiC ensures its stability and mediates the incorporation of CSA into the CRL^CSA complex. Our findings suggest a hand-over mechanism in which TRiC provides properly folded CSA to DDB1, which is crucial to enable the formation of the CRL^CSA complex and its nuclear localization. Interfering with the TRiC–CSA interaction, either by disturbing or strengthening it, lowers the levels of functional CSA in the nuclear CRL^CSA complex and results in impaired recovery of RNA synthesis and decreased cell viability upon UV-C-induced DNA damage. Thus, we uncover CSA as a TRiC substrate and reveal a role for the TRiC chaperonin in regulating CSA-dependent TC-NER.

CSA has been shown to stably interact with DDB1[6]. However, our iBAQ analysis suggests that approximately 15% of the CSA protein pool is not bound by DDB1 (Fig. 2b). This fraction of CSA is likely unstable and/or improperly folded and therefore bound by TRiC. Consistently, pulldowns of CRL^CSA revealed that TRiC preferentially binds CSA that is not associated with the CRL complex (Fig. 1e). From our iBAQ analysis, a (DDB1-free) CSA to TRiC subunit ratio of ~1:2 can be inferred. As every TRiC complex contains two copies of each of the eight subunits, this stoichiometry may suggest a model in which one CSA protein is encapsulated per TRiC complex. Interestingly, this model differs from the proposed encapsulation mode for the TRiC substrate tubulin, for which two molecules were shown to bind the complex simultaneously[35]. This suggests that TRiC employs different methods of substrate binding and folding. To fully understand the constitution and conformation of TRiC in complex with CSA, a more detailed structural analysis would be required.

Our results suggest that TRiC interacts with CSA through its WD40 domain, thereby regulating CSA stability. Interestingly, TRiC has been described to regulate the folding and stability of several other WD40 domain-containing proteins[25,36–42]. For instance, TRiC is required to maintain functional TCAB1, a co-factor of telomerase. Loss of TRiC leads to mislocalization of telomerase and a failure to elongate telomeres[25]. Importantly, TCAB1 mutations found in patients with dyskeratosis congenita (DC), which is a stem cell disease caused by defects in telomere maintenance[43], were shown to disrupt TRiC-mediated TCAB1 folding, providing clinical relevance to TRiC's role in stabilizing this protein. Mutations in CSA have been mostly linked to CS[12]. All types of mutations (missense, nonsense, frameshift, splicing mutations, as well as large deletions) have been detected in CS patients[44]. With the exception of the missense mutations, most mutations likely lead to the production of a truncated and/or non-functional CSA protein, providing a plausible explanation for the cause of CS. Interestingly, the majority of the missense mutations were found in the seven WD motifs that form the WD40 domain[16,44]. Here we show that three of these patient mutations lead to protein instability, resulting in increased TRiC binding and consequently a loss of functional CRL^CSA-bound CSA in the nucleus. Whether the other reported disease-causing missense mutations similarly impact TRiC-mediated folding and stabilization of CSA remains to be established.

DNA repair defects are a major source of genomic instability. Given that TRiC by affecting CSA stability contributes to TC-NER, it may play an important role in preserving genome stability following UV damage. Whether TRiC generally preserves genome stability by affecting DNA damage repair pathways other than TC-NER is not clear and may require the identification of additional, yet to be identified substrates. However, in support of such a scenario, it was shown that TRiC regulates the stability of the p53 tumor suppressor protein that is involved in genome stability maintenance[45]. In addition, TRiC was found to regulate the folding and stability of the WD40 domain-containing CDC20 protein[36,46], which is a member of the anaphase-promoting complex. CDC20 controls cell division and genome integrity and has been implicated in cancer[47]. Thus, TRiC likely affects genome stability maintenance by facilitating the folding of proteins other than CSA. Future endeavors may shed light on how misregulation of TRiC generally affects genome instability and contributes to diseases such as cancer[48]. Such work may also provide potential targets for diagnostics and therapeutics for pathological conditions associated with genome instability, such as cancer and aging-related diseases.

## Methods

**Cell culture.** Cells were cultured in DMEM (Invitrogen) supplemented with 10% fetal bovine serum (FBS; Bodinco BV) and penicillin/streptomycin (Sigma). The following cell lines were used: U2OS (ATCC), CS3BE-SV40 (GM01856; Coriell Institute), CS3BE-hTert (GM01856; Coriell Institute), and VH10-hTert.

**Generation of stable cell lines.** Constructs encoding CSA-FLAG were established by cloning CSA cDNA (extended with a FLAG-tag by PCR) into pENTR4 (Invitrogen). GFP-tagged constructs were made by cloning CSA WT or CSA 8M, which was created by site-directed mutagenesis using the QuickChange site-directed mutagenesis kit (Agilent), into pENTR1A-GFP-N2 (Addgene). CSA constructs harboring single amino-acid substitutions E103A, F120A, K122A, R164A, K247A, K292A, K292A + K293A, and R354A and a C-terminal 10× -His-tag were created by PCR and cloned into pDONR221. Constructs were subsequently transferred to pLenti6.3 V5-DEST (pENTR4, pENTR1A-GFP) or pLenti4 V5-DEST (pDONR221) by Gateway LR Clonase II Enzyme Mix (Invitrogen). Lentivirus was produced using the pCMV-VSV-G, pMDLg-RRE and pRSV-REV plasmids (Addgene) and used to infect cells with Polybrene® (Sigma). Stable integrands were obtained after selection in medium containing blasticidin (ThermoFisher Scientific) (pLenti6.3) or zeocin (Invitrogen) (pLenti4).

U2OS Flp-In/T-REx cells, which were generated by Professor J. Parvin using the Flp-In™/T-REx™ system (Thermo Fisher Scientific), were a gift of Dr. S. Pfister. These cells were co-transfected with pLV-U6g-PPB containing an antisense guide RNA targeting the CSA/ERCC8 gene (5-CCAGACTTCAAGTCACAAAGTTG-3) from the LUMC/Sigma-Aldrich sgRNA library together with an expression vector encoding Cas9-2A-GFP (pX458; Addgene #48138). Transfected U2OS Flp-In/T-REx cells were selected on puromycin for 3 days, plated at low density, after which individual clones were isolated. Knockout of CSA and the absence of Cas9 integration/stable expression in the isolated clones was verified by western blot analysis. The neomycin resistance gene in pcDNA5/FRT/TO-Neo (Addgene #41000) was replaced with a puromycin resistance gene to generate pcDNA5/FRT/TO-Puro. A fragment spanning GFP-NLS or GFP-N1 (Clontech) was inserted in this vector to create pcDNA5/FRT/TO-GFP-NLS-Puro and pcDNA5/FRT/TO-GFP-N1-Puro, respectively. CSA WT or CSA ΔN (lacking the first 21 amino acids) were amplified by PCR (primers: CSA WT 5′-CACAATGCTAGCGCCACCATGCTGGGGTTTTTGTCCG-3′ and 5′-GCATGGTGAAC TACCGGTGCTCCTTCTTCATCACTGCTG-3′, CSA ΔN 5′-CTAGTAGAATTCATCGGACG CTAG CATGGAGTCAACACGGAGAGTTTTGG-3′ and 5′-GCACCGACGACCTAGG CAGGATCCAGACTTCAAGTCACAAAG-3′) and inserted into pcDNA5/FRT/TO-GFP-N1-Puro. One of the CSA knockout clones was subsequently used to stably express GFP-NLS, CSA-GFP WT or CSA–GFP ΔN by co-transfection of pCDNA5/FRT/TO-Puro plasmid encoding these CSA variants (2 µg), together with pOG44 plasmid encoding the Flp recombinase (0.5 µg). After selection on puromycin, single clones were isolated and expanded. Isolated U2OS CSA knockout clones stably expressing CSA-GFP WT or CSA–GFP ΔN were selected based on their equal and near-endogenous expression levels.

**Generation and expression of CSA patient mutants.** CSA cDNA was cloned into pEGFP-N2 (Addgene). Mutations A160T, A205P, and D266G were created by site-directed mutagenesis using the QuickChange site-directed mutagenesis kit (Agilent). Plasmids were transfected using Lipofectamine® 2000 (Invitrogen) in Opti-MEM™ (Gibco) containing 10% FBS. Twenty-four hours after transfection, cells were used for GFP-pulldown or fluorescence microscopy.

**RNA interference.** Proteins were depleted by two sequential transfections with 40 nM siRNA (Dharmacon, GE Healthcare) using Lipofectamine® RNAiMAX (Invitrogen) in Opti-MEM™ (Gibco) containing 10% FBS. The following siRNAs were used:

5′-CGUACGCGGAAUACUUCGA-3′ (Luciferase);
5′-GCAAGGAAGCAGUGCGUUAUU-3′ (TCP1-1);
5′-GACCAAAUUAGACAGAGAUU-3′ (TCP1-2);
5′-GAACUGAGUGACAGAGAAAUU-3′ (CCT4-1);
5′-GUGUAAAUGCAGUGAUGAAUU-3′ (CCT4-2);
5′-GCAAAUACAAUGAGAACAUUU-3′ (CCT5-1);
5′-CAACACAAAUGGUUAGAAUUU-3′ (CCT5-2);
5′-CUGACAACUUUGAAGCUUUUU-3′ (CCT7-1);
5′-GGCAAUUGUUGAUGCUGAGUU-3′ (CCT7-2);
5′-UGAUAAUGGUGUUGUGUUUUU-3′ (DDB1-1);
5′-AGAGAUUGCUCGAGACUUUUU-3′ (DDB1-2).

**UV-C irradiation.** UV damage was induced using a 254-nm TUV PL-S 9W lamp (Philips).

**Treatment with TRiC inhibitor.** Medium supplemented with 2.5 mM 2-[(4-chloro-2$\lambda^4$,1,3-benzothiadiazol-5-yl)oxy]acetic acid (Vitas-M Laboratory Ltd., via MolPort-002-507-960) was added to attached cells in six-well plates every 24 h during 72 h.

**Western blotting.** Proteins were separated in 4–12% Bis-Tris NuPAGE® gels (Invitrogen) or Criterion™ gels (Bio-Rad) in MOPS (Life Technologies). For the detection of (endogenous) CSA by the Abcam rabbit CSA antibody, hand casted 10% or 13% acrylamide gels were used and electrophoresis was performed in a Tris-Glycine-SDS buffer. Separated proteins were blotted onto PVDF membranes (Millipore), which were incubated with the following primary antibodies: rabbit α-FLAG (Sigma, F7425; 1:2000); mouse α-Tubulin (Sigma, T6199; 1:5000); mouse α-GFP (Roche, #11814460001; 1:1000); mouse α-RNAPIIo (Abcam, ab5408; 1:1000); goat α-DDB1 (Abcam, ab9194; 1:1000); rabbit α-CSA/ERCC8 (Abcam, ab137033; 1:1000); rabbit α-H3 (Abcam, ab1791; 1:5000); rabbit α-CSB/ERCC6 (Santa Cruz Biotechnology, sc-25370; 1:1000); goat α-CSB/ERCC6 (Santa Cruz Biotechnology, sc-10459; 1:1000); mouse α-CCT4 (Santa Cruz Biotechnology, sc-137092; 1:500); rabbit α-CUL4A (Bethyl Laboratories, A300-739A; 1:500); mouse α-TCP1 (Abnova, H00006950-M01; 1:1000); mouse α-CCT5 (Abnova, H00022948-M01; 1:500); mouse α-CCT7 (Abnova, H00010574-M01; 1:500). Protein bands were visualized using the Odyssey® Imaging System (LI-COR) after incubation with CF™ dye labeled secondary antibodies (Sigma; 1:10,000), or detected by the ECL™ Prime Western Blotting system (GE Healthcare) following incubation with Horseradish Peroxidase-conjugated secondary antibodies (Dako; 1:5000).

**Immunoprecipitations and pulldowns.** Cells were lysed in IP buffer (30 mM Tris pH 7.5, 150 mM NaCl, 2 mM MgCl₂, 0.5% Triton X-100, protease inhibitor cocktail (Roche)) during 1 h at 4 °C. The supernatant obtained by centrifugation is referred to as the soluble fraction, while the solubilized chromatin fraction was prepared by resuspension of the pellet followed by 1–2 h of incubation in IP buffer containing 250 U/mL Benzonase® Nuclease (Novagen). Samples were subsequently incubated with the indicated antibody for immunoprecipitation during 2–4 h.

For immunoprecipitation of proteins from total cell extracts, cells were directly lysed in IP buffer supplemented with 250 U/mL Benzonase® nuclease and the desired antibody. Protein complexes were pulled down during 1–2 h incubation with Protein A agarose beads (Millipore). GFP-tagged proteins were precipitated using GFP-Trap®_A beads (Chromotek), while FLAG-tagged proteins were precipitated using ANTI-FLAG® M2 Affinity Agarose Gel (Sigma). For tandem purification, proteins were eluted from the beads by addition of 3× FLAG peptide (Sigma). For subsequent analysis by western blotting, proteins were eluted by boiling of the beads in Laemmli-SDS sample buffer.

**Determination of overall protein levels by western blotting.** For detection of overall protein levels, whole-cell extracts were prepared by lysis in 5 µL IP buffer (30 mM Tris pH 7.5, 150 mM NaCl, 2 mM MgCl₂, 0.5 % Triton X-100, protease inhibitor cocktail (Roche)) per 100,000 cells during 10 min at room temperature. Equal volumes of Laemmli-SDS sample buffer were added and the samples were heated at 95 °C for 10 min prior to western blot analysis.

**Fluorescence microscopy.** Cells were grown on glass coverslips and subjected to the indicated treatments. Cells were washed with PBS and fixed with 2% formaldehyde (Sigma) in PBS. For nuclear staining, cells were permeabilized in 0.25% Triton X-100 (Sigma) and incubated with DAPI (Sigma). Images were acquired on a Zeiss AxioImager D2 widefield fluorescence microscope equipped with ×40, ×63, and ×100 PLAN APO (1.4 NA) oil-immersion objectives (Zeiss) and an HXP 120 metal-halide lamp used for excitation. Images were recorded using ZEN 2012 software and analyzed in ImageJ (https://imagej.nih.gov/ij/).

**Identification of CSA-interacting proteins.** For stable isotope labeling of amino acids in culture (SILAC), cells were grown in DMEM containing 10% dialyzed FBS (Gibco), 10% GlutaMAX (Life Technologies), penicillin/streptomycin (Life Technologies), unlabeled L-arginine-HCl and L-lysine-HCL or $^{13}C_6,^{15}N_4$L-arginine-HCl and $^{13}C_6,^{15}N_2$L-lysine-2HCL (Cambridge Isotope Laboratories), respectively. FLAG

and CSA-FLAG complexes were pulled down from total cell extracts with ANTI-FLAG® M2 Affinity Gel (Sigma) and extensively washed. Bound proteins were eluted with FLAG peptide (0.2 mg/mL in PBS), separated in SDS-PAGE gels and visualized with Coomassie (SimplyBlue; Invitrogen). SDS-PAGE gel lanes were cut into 2-mm slices and subjected to in-gel reduction with dithiothreitol, alkylation with iodoacetamide (98%; D4, Cambridge Isotope Laboratories) and digestion with trypsin (sequencing grade; Promega). Nanoflow liquid chromatography tandem mass spectrometry (LC-MS/MS) was performed on an 1100 series capillary liquid chromatography system (Agilent Technologies) coupled to a Q-Exactive mass spectrometer (Thermo Scientific) operating in positive mode. Peptide mixtures were trapped on a ReproSil C18 reversed phase column (1.5 cm × 100 µm) at a rate of 8 µL/min, separated using a linear gradient of 0–80% acetonitrile (in 0.1% formic acid) during 60 min at a rate of 200 nL/min using a splitter. The eluate was directly sprayed into the electrospray ionization (ESI) source of the mass spectrometer. Spectra were acquired in continuum mode; fragmentation of the peptides was performed in data-dependent mode. Mass spectrometry data were analyzed with MaxQuant software (version 1.1.1.25).

**LFQ and cross-linking mass spectrometry.** LFQ, stoichiometry estimation, and cross-linking mass spectrometry were performed essentially as described previously[20,21]. Briefly, GFP immunoprecipitations for LFQ and stoichiometry analysis were performed in triplicate using ChromoTek GFP-Trap beads or control non-GFP beads and 2 mg of whole-cell lysate collected in a 1% NP-40 whole-cell lysis buffer. After protein incubation, two washes were performed with 1 M NaCl and 1% NP-40, followed by additional washes with PBS. Reduction and alkylation were performed in-solution, and samples were digested with trypsin overnight. Tryptic peptides were separated over a 120 min gradient from 7 to 32% acetonitrile with 0.1% formic acid and measured on a Thermo Q-Exactive mass spectrometer. Identification and quantification of peptides were performed using MaxQuant version 1.5.1.0[49]. Relative stoichiometries were calculated by normalizing each protein by iBAQ value against the bait protein (CSA).

For cross-linking mass spectrometry, two independent experiments were conducted. Protein purifications and mass spectrometry analysis were essentially the same as stated above, with exceptions noted below. First, after washes, we cross-linked immunoprecipitated complexes on-bead for 1 h at room temperature using 1 mM BS3 (bis(sulfosuccinimidyl)suberate) in 50 mM borate buffered saline. Cross-linking was quenched with 100 mM ammonium bicarbonate for ten minutes and sample preparation for mass spectrometry was continued as previously, including reduction, alkylation, and digestion. Samples were measured on either a Thermo QExactive or a Thermo Fusion as above, but over a 4 h 7–37% acetonitrile gradient with charge 2+ or lower masses excluded from fragmentation. Cross-linked peptides were identified using pLink[22] with an FDR of 0.05. Identified cross-links were further filtered to remove matches were either peptide was not ≥5 or ≤40 amino acids in length and with an e-value for the spectral match of ≤0.0001. All identified cross-links in any experiment meeting these criteria were combined for further analysis. Cross-linking data were structurally validated using a TRiC homology model where each subunit was produced using Phyre2 and aligned onto the eukaryotic TRiC in Chimera (PDB: 4V94[50,51]). In cases where a cross-linked residue was not resolved in the structure, the nearest structurally resolved residue in the protein sequence was used for modeling. All structural images were produced in UCSF Chimera, and cross-link distance analysis was performed using XlinkAnalyzer[52,53]. Accessible interaction space was modeled using DisVis[23] and human CSA (PDB: 4A11[6]).

**RNA synthesis recovery assay.** Cells were seeded in 96-well plates, transfected with siRNAs (see above) and after 48 h irradiated with UV-C (10 J/m²), and incubated for different time-periods (0–30 h) to allow RNA synthesis recovery. RNA was labeled for 1 h in medium supplemented with 1 mM EU (Click-iT® RNA Alexa Fluor® 594 Imaging Kit, Life Technologies) according to the manufacturer's instructions. Imaging was performed on an Opera Phenix confocal High-Content Screening System (Perkin Elmer, Hamburg, Germany) equipped with solid state lasers. General nuclear staining (DAPI) and Alexa 594 were serially detected in nine fields per well using a ×20 air objective. Three independent experiments were analyzed using a custom script in the Harmony 4.5 software (Perkin Elmer) in which nuclei were individually segmented based on the DAPI signal. RNA synthesis recovery was determined by measuring the mean Alexa 594 intensity of all nuclei per well.

**DNA synthesis repair assay.** Cells were seeded on coverslips and transfected with siRNA (see above). After 48 h, the cells were UV-C irradiated (20 J/m²) and subsequently DNA was labeled for 3 h in medium supplemented with 1 µM of EU (Click-iT® DNA Alexa Fluor® 488 Imaging Kit, Life Technologies) according to the manufacturer's instructions. DNA synthesis repair was quantified by determining fluorescence intensities for >20 cells with ImageJ software of images obtained with a Zeiss LSM700.

**UV and Illudin S survival assays.** Cells were seeded at low density and UV-C irradiated at different doses or treated with 300, 600, and 1000 pg/mL Illudin S (Santa Cruz; sc-391575) for 72 h. After 11–14 days of incubation, cells were

washed with 0.9% NaCl and stained with methylene blue. Colonies of >20 cells were scored.

**Cell viability (alamarBlue) assay.** Cells were seeded in 96-well plates, transfected with siRNAs (see above) and after 48 h irradiated with UV-C (10 J/m$^2$). Alamar-Blue® (Life Technologies) was added and fluorescence was measured 72 h later according to the manufacturer's instructions.

**Data availability**. The data sets generated and analyzed during the current study have been deposited to the ProteomeXchange Consortium via the PRIDE partner repository with the data set identifiers PXD008863 and PXD008868. Other relevant data generated during the current study are available from the corresponding authors on reasonable request.

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

## Acknowledgements

We thank Professor F.-U. Hartl and colleagues for constructive discussions on our work. We also thank Professors R. Houben and M. Rabelink for providing gRNAs from the LUMC/Sigma-Aldrich library, and Professor J. Parvin and Dr. S. Pfister for sharing U2OS Flp-In/T-REx cells. This work was financially supported by an ERC-Consolidator grant (50364, H.v.A.), an ERC-Advanced grant (340988, W.V. and A.P.), grants from the Netherlands Organisation for Scientific Research (NWO) (ALWOP-143, A.P.; VIDI-700.55.425, N.S.P.; VIDI-016.161.320, M.S.L.; TOP Talent-021.002.024, E.M.M.), the EU FP7 ITN project DevCom (607142; M.M. and M.G.V.), the Netherlands Toxicogenomics Centre (05060510; L.H.M.), Leiden University (Research Profile Bioscience: The Science Base of Health'; L.H.M. and N.S.P.), and the NWO Gravitation program Cancer Genomics Netherlands (W.V.). The Cancer Treatment Screening Facility (CTSF) is funded by the Daniel den Hoed Foundation (M.E.v.R. and P.J.F.).

## Author contributions

A.P. and M.D. generated plasmids and performed fluorescence microscopy experiments, coIPs, pulldowns, protein stability, survival assays and mass spectrometry. M.M., M.B., and M.V. performed and analyzed xIP-MS experiments. E.M.M., M.G.V., and N.S.P. designed CSA 8M, generated CSA 8M constructs and stable cell lines and performed survival assays. P.J.F. and M.E.v.R. measured and analyzed RNA synthesis recovery. M.S.L. and Y.v.d.W. designed and created CSA ΔN, generated CSA knockout cells and performed experiments related to CSA ΔN. A.P., M.D., W.V., and H.v.A. wrote the manuscript. L.H.M., N.S.P., W.V., and H.v.A. supervised the project.

## Additional information

**Competing interests:** The authors declare no competing interests.

