## [Peer Review File · Nature Communications]

Reviewers' Comments:

Reviewer #1 (Remarks to the Author):

The authors found TRiC to interact directly with CSA. Knockdown of TRiC subunits resulted in a decrease in the amount of CSA, a reduction in resumption of RNA synthesis after UV irradiation, and hypersensitivity to UV light. DDB1 knockdown and a CSA mutant that increased binding to TRiC showed a decrease in the formation of CRL^{CSA} complex and an increase in cytoplasmic localization of CSA. Cells expressing the CSA mutant also showed reduced resumption of RNA synthesis after UV irradiation and hypersensitivity to UV light. In addition, three missense mutation of CSA from patients with Cockayne syndrome did not bind to DDB1, but more to TRiC. From these results, the authors claimed that TRiC functions in CSA folding and stabilization, assembly of the CRL^{CSA} complex, and the development of Cockayne syndrome.

The findings are novel and would be interesting to many readers of the journal.

However, there are several points that should be addressed to strengthen their claims.

1. By western blotting after immunoprecipitation of CSA (Fig. 1, 4 and 5), both TCP1 and CRL components were detected. To clarify the role of TRiC, the authors should indicate whether TCP1 only binds to DDB1-free CSA by performing gel filtration of the immunoprecipitated samples or immunoprecipitation of TCP1 etc.. If TCP1 interacts with DDB1-bound CSA, the interpretation of the results is changed. It is also necessary to add images of coprecipitated other TRiC components.
2. Knockdown experiments alone are insufficient to claim the functions of TRiC in the formation of CRL^{CSA}. Experiments using functional deficient TRiC and/or purified TRiC are necessary. Furthermore, it would be better to analyze a CSA mutant that do not interact with TRiC. A mutant of the 4 Lys residues (34, 85, 167, and 212) of CSA does not bind to TRiC?
3. Knockdown of TRiC components reduced the amount of CSA in the extracts (Fig. 3a, 3b, and 3c). It should be shown that the amounts of CRL^{CSA} in the knockdown cells is also reduced as compared to those in the control cells. This is the basis for the interpretation of Fig. 6a and 6c.
4. In Fig. 5c, a certain amount of CSA-GFP 8M was incorporated into the CRL complex. The amount of CRL^{CSA-FLAG 8M} (and CRL^{CSA-GFP 8M}) should be

compared with that of endogenous CRL^{CSA}. This is the basis for the interpretation of Fig. 6b and 6d.

5. Does knockdown of TCP1 have no effect on RNA synthesis? This should be mentioned in the text. RNA synthesis was measured at two points after UV irradiation (Fig. 6a and 6b). It is better to add points at 16 or 18 h and 30 or 32 h after UV irradiation.

6. The UV doses (20, 40, and 60 J/m²) used in Fig. 6C are relatively high. This needs to be explained.

Reviewer #2 (Remarks to the Author):

Review of manuscript NCOMMS-17-19665

The manuscript entitled “TRiC controls transcription resumption after UV damage by regulating Cockayne Syndrome protein A” by Pines and colleagues provides a detailed analysis of the interaction between TRiC and CSA and its functional relevance for TC-NER. The manuscript is well written and the experimental data of high quality. The authors show that CSA is destabilized upon depletion of the TRiC subunit TCP1. In contrast, destabilisation of the CRL-CSA complex by depletion of DDB1 increases the association of CSA with TRiC. Likewise, mutation of the WD40 domain in CSA causes increase association with TRiC and reduced incorporation in the CRL complex. Based on these data the authors conclude that TRiC folds and hands over CSA to the CRL complex and that this is important for TC-NER due to the UV sensitivity caused by TCP1 depletion or mutation of the WD40 folds in CSA. There are several weaknesses of these conclusions that must be addressed before publication.

Major concerns:

1. The hand-over model seems counter-intuitive, because TRiC is cytoplasmic and CSA localized to the nucleus. The authors should provide evidence for this model e.g. by showing that CSA is incorporated in the CRL complex in the cytoplasm before nuclear import. Alternatively, the authors should provide alternatives e.g. that folding of CSA by TRiC in the cytoplasm and incorporation of CSA in the CRL complex are separate events although the latter may depend on the first. Does depletion of TCP1 lead to decreased binding to DDB1 as predicted by the model?
2. Page 10, the authors conclude that TRiC folds the 8M mutant. What is the evidence for this and that the 8M mutant is misfolded in the first place?
3. Page 11, the authors conclude “that TRiC, by regulating CSA stability and incorporation into the CRLCSA complex, promotes TC-NER and protects cells against UV-induced damage”. The reduced viability after UV irradiation of cells depleted for TCP1 (Figure 6C) could be due to

another substrate of TRiC. Since incorporation of CSA into the CRL complex is only mildly reduced by depletion of TRiC, the authors should validate their model by testing if mild overexpression of CSA will complement the UV sensitivity caused by depletion of TCP1.

Minor question:

1. Do the FLAG- and GFP-tagged versions of CSA complement?

Response letter

Revised manuscript “*TRiC controls transcription resumption after UV damage by regulating Cockayne Syndrome protein A*” by Pines, Dijk *et al.*

We thank the editor for providing us the opportunity to respond to the comments of the reviewers, and allow the submission of a revised manuscript. We are also grateful to the reviewers for their constructive comments and unanimous appreciation of the originality and significance of our manuscript. All reviewers agree that the manuscript discloses important mechanistic insight on TC-NER, thereby improving our understanding of how this DNA repair mechanism prevents disease development.

Our point-by-point response to the reviewer’s comments can be found below in red marked text. All changes that we made to the text in our manuscript are also marked in red. Several panels with new results were added to the figures as indicated in our response.

Reviewer #1:

1. By western blotting after immunoprecipitation of CSA (Fig. 1, 4 and 5), both TCP1 and CRL components were detected. To clarify the role of TRiC, the authors should indicate whether TCP1 only binds to DDB1-free CSA by performing gel filtration of the immunoprecipitated samples or immunoprecipitation of TCP1 etc.. If TCP1 interacts with DDB1-bound CSA, the interpretation of the results is changed.

The reviewer suggested us to perform a gel filtration of immunoprecipitated CSA samples to verify whether DDB1-free CSA interacts mainly with TRiC. Unfortunately, gel filtration is not a feasible option due to the small amount of CSA that is immunoprecipitated, which is in the order of nano-grams. Moreover, gel filtration is not sufficiently discriminative to resolve similarly large molecular complexes such as TRiC and CRL associated with its regulator CSN (COP9 signalosome). Instead, we addressed this issue with a more discriminative approach involving a tandem pulldown. We first expressed GFP-DDB1 in cells stably expressing FLAG-tagged CSA and performed a FLAG pulldown. As expected, we were able to pulldown both the

CRL (CUL4A and DDB1) and TRiC (CCT7 and CCT4) complexes with FLAG-CSA (Figure 1e). We then performed a GFP pulldown in which we recovered CRL, but not TRiC components (Figure 1e). This new result corroborates our model of mutually exclusive binding of CSA to either DDB1 or TRiC. However, we cannot exclude the formation of a transient TRiC-CRL^{CSA} complex in which TRiC and DDB1 coexist at the moment of the proposed handover of CSA from TRiC to DDB1 in the CRL^{CSA} complex.

It is also necessary to add images of coprecipitated other TRiC components

In the newly added panels in Figure 1 (Figure 1d and Figure 1e) we have included immunostainings for additional TRiC components: CCT4, CCT5 and CCT7. This revealed an interaction between CSA and these TRiC components, corroborating our CSA pulldown/mass spectrometry findings from Figure 1a.

2. Knockdown experiments alone are insufficient to claim the functions of TRiC in the formation of CRLCSA. Experiments using functional deficient TRiC and/or purified TRiC are necessary.

It would indeed be interesting to study TRiC's function in CRL^{CSA} complex formation *in vitro* using purified TRiC. However, this would not only require the production of purified functionally deficient TRiC complexes, but also functional TRiC and CRL^{CSA} complexes, which we believe is not realistic and beyond the scope of this study. However, as an alternative to knockdown experiments to study the functions of TRiC, we have now used a chemical inhibitor against TRiC. Following treatment of cells with this inhibitor, which renders TRiC functionally deficient by inhibiting TCP1, we observed a clear reduction in the amount of CSA (Figure 3d) (similar to that observed after TCP1 knockdown, Figure 3a,b), corroborating our finding that TRiC's activity is important for maintaining CSA stability and, consequently, the formation of CRL^{CSA} complex.

Furthermore, it would be better to analyze a CSA mutant that do not interact with TRiC. A mutant of the 4 Lys residues (34, 85, 167, and 212) of CSA does not bind to TRiC?

The analysis of a CSA mutant that cannot interact with TRiC would indeed be interesting. The reviewer suggested to analyse a mutant of the 4 Lysine residues as identified in CSA by XL-MS. This approach showed multiple links between CSA and specific subunits of the TRiC complex, revealing the encapsulation of CSA in the chaperonin complex cavity. Importantly, this procedure provides molecular distance restraints by identifying spatially proximate lysine residues within a single protein (intra-links) or between multiple proteins (inter-links) that have been covalently linked by a homo-bi-functional crosslinking reagent that reacts with the amino group of lysine residues. However, it does not provide any information on specific residues that mediate the interaction. Concerning the 4 Lysine residues (34, 85, 167, and 212), we can only conclude that they are spatially proximate to TRiC. Finally, many mutations in CSA may either not affect the interaction with TRiC (just like the 8 mutations we already tested individually, Supplementary Figure 4), or enhance the interaction with TRiC due to folding problems (like the CSA 8M mutant, Figure 5), making it difficult if not impossible to obtain a CSA mutant that does not interact with TRiC.

3. Knockdown of TRiC components reduced the amount of CSA in the extracts (Fig. 3a, 3b, and 3c). It should be shown that the amounts of CRLCSA in the knockdown cells is also reduced as compared to those in the control cells. This is the basis for the interpretation of Fig. 6a and 6c.

Several approaches have been used to address whether the amount of CSA that is assembled into the CRL complex would be reduced upon TRiC knockdown. We performed pulldowns of endogenous DDB1 and pulldowns of GFP-DDB1 from U2OS cells, but failed to detect co-precipitated CSA (data not shown). In addition, we performed pulldowns of GFP-DDB1 from CSA-deficient CS3BE-SV40 patient cells (over)expressing FLAG-tagged CSA, but even under these conditions we were unable to detect co-precipitated CSA, while the presence of CUL4 was evident (Figure R0; for reviewers only). It should be noted that a large collection of different CRL (CUL4-RBX1-DDB1) complexes exists with as many as 90 different adapter proteins that associate with DDB1 to determine specificity (He *et al.*, *Genes & Dev.*, 2006). Therefore, only a very small minority of the total pool of CRL complexes will contain CSA. This likely explains why we cannot detect CSA following immunoprecipitation/pulldown of DDB1. To circumvent this problem and address the

reviewer's issue, we used an alternative approach. We overexpressed CSA in cells depleted of TCP1 to examine if this could rescue the loss of CSA and the reduced availability of active CRL^{CSA} complexes. We indeed found that overexpression of CSA in the TCP1-depleted partially rescued the UV sensitivity of these cells, suggesting that TCP1/TRiC regulates the availability of functional CRL^{CSA} complexes by promoting proper folding and stability of CSA (Supplementary Figure 6b; see also response to point 3 of reviewer 2).

Figure R0. GFP and GFP-DDB1 were transiently expressed and pulled down from CS3BE-SV40 cells which stably expressed CSA-FLAG and were treated with the indicated siRNAs. Protein levels were determined by Western blot analysis of the indicated proteins.

4. In Fig. 5c, a certain amount of CSA-GFP 8M was incorporated into the CRL complex. The amount of CRLCSA-FLAG 8M (and CRLCSA-GFP 8M) should be compared with that of endogenous CRLCSA. This is the basis for the interpretation of Fig. 6b and 6d.

We do not fully understand this request, as it is impossible to compare CSA-FLAG 8M and CSA-GFP 8M to the endogenous amount of CSA, since these experiments were

performed in CSA-deficient patient cells in which no CSA protein can be detected. It should be noted, however, that these CSA-deficient patient cells were carefully selected for equal expression of CSA-GFP WT and CSA-GFP 8M (see input signals in Figure 5c). Under these conditions, CSA-GFP 8M shows an enhanced interaction with TCP1/TRiC (Figure 5c). Moreover, CSA-GFP WT rescues the TC-NER defect of CSA-deficient cells, whereas CSA-GFP 8M is not capable of doing so (Figure 6b,d).

5. Does knockdown of TCP1 have no effect on RNA synthesis? This should be mentioned in the text.

We have added a panel with new results to Supplementary Figure 4 (Supplementary Figure 4a). The new results show that knockdown of TCP1 does not affect basal levels of RNA synthesis. We have also mentioned this in the text of our revised manuscript.

RNA synthesis was measured at two points after UV irradiation (Fig. 6a and 6b). It is better to add points at 16 or 18 h and 30 or 32 h after UV irradiation.

We have performed new experiments (in quadruplicate) with additional time points after UV irradiation as requested. These results are presented in a new panel of Figure 6 (Figure 6a) and corroborate our previous finding that loss of TCP1 impairs TC-NER.

6. The UV doses (20, 40, and 60 J/m²). used in Fig. 6C are relatively high. This needs to be explained.

The applied UV doses for the cell viability assays in Figure 6c are indeed relatively high when compared to those used in classical clonogenic survival assays in Figure 6d. These doses are however similar to the doses typically used for this assay (Lee *et al.*, Cancer Res., 2005; Ghodgaonkar *et al.*, DNA Repair, 2008; Fan *et al.*, Mol. Cell, 2010). To investigate the effect of TRiC knockdown on UV sensitivity, we could not use the classical clonogenic UV survival assay, as this is based on long term culturing (to allow colony formation) after a certain single dose of UV. Colony formation, even without UV irradiation, is affected by extensive culturing of cells in the absence of TRiC, whose components are essential for cell viability (Winzeler *et al.*, Science 1999; Spies *et al.*, Trends Cell Biol., 2004; Hart *et al.*, Cell, 2015), and therefore confound

the outcome. To circumvent this issue, we have performed a short-term assay to monitor UV sensitivity. To measure short-term effects of UV on cell viability high UV doses are required.

Reviewer #2

1. The hand-over model seems counter-intuitive, because TRiC is cytoplasmic and CSA localized to the nucleus. The authors should provide evidence for this model e.g. by showing that CSA is incorporated in the CRL complex in the cytoplasm before nuclear import. Alternatively, the authors should provide alternatives e.g. that folding of CSA by TRiC in the cytoplasm and incorporation of CSA in the CRL complex are separate events although the latter may depend on the first.

Our hand-over model suggests that correctly folded CSA, provided by the cytoplasmic TRiC chaperonin (Frydman *et al.*, EMBO. J., 1992), will be delivered to its receptor (DDB1) in the CRL complex, after which this complex is quickly translocated to the nucleus. This model is further substantiated by the notion that when the receptor protein (DDB1) is depleted the nuclear import of CSA is significantly affected, as shown in Figure 4a and 4b. In support of this, we now show that CSA, which lacks the first 21 N-terminal amino acids required for interaction with DDB1 (Fischer *et al.*, Cell, 2013), remains mostly associated with TRiC/TCP1, likely as a consequence of impaired hand-over (see new results in Figure 6e). In other words when the hand-over to its preferred complex partner (DDB1) is interrupted by DDB1 depletion or by impairing the CSA-DDB1 interaction, CSA remains associated with cytoplasmic TRiC. In order to provide further evidence for this model, we performed cellular fractionation experiments using cells expressing CSA-GFP. Pulldown of CSA-GFP from cytoplasmic fractions showed that CSA strongly interacts with DDB1 (see new result in Supplementary Figure 3a,b), which is agreement with a hand-over of CSA from cytoplasmic TRiC to DDB1 in the CRL complex.

Does depletion of TCP1 lead to decreased binding to DDB1 as predicted by the model?

Please see our response to point 3 of reviewer 1.

2. Page 10, the authors conclude that TRiC folds the 8M mutant. What is the evidence for this and that the 8M mutant is misfolded in the first place?

We nowhere state that TRiC folds the CSA 8M mutant. In fact, on page 11 (formerly page 10) we conclude that TRiC folds CSA and not mutant CSA: “*This corroborates our conclusion that cytoplasmic TRiC provides properly folded CSA to DDB1 for incorporation into CRL^{CSA} complexes and subsequent translocation into the nucleus*”.

We have no evidence that CSA 8M is misfolded. Several residues in CSA were identified as potential substrate-interacting residues: Glu103, Phe120, Lys122, Arg164, Lys247, Lys292, Lys293, Arg354; see Figure 5a,b). These residues were selected using the following criteria: the side-chains should be probable candidates for interaction (e.g. charged or big hydrophobic side-chains), the side-chains should be interacting with or otherwise be very near substrate peptides in a superposition of CSA with co-crystal structures of WD40 proteins with peptides, and the side-chains should not be involved in any interaction that is likely to be important for the stability of the structure of the protein itself. Indeed, we found that the expression of CSA carrying mutations of either of these residues, fully complemented the UV sensitivity phenotype of CSA-deficient patient cells. This suggest that mutations of either of these residues alone does not impair CSA’s function in the UV damage response. We therefore decided to combine these mutations in the CSA 8M mutant. This mutant, however, failed to complement the UV sensitivity and TC-NER defect of CSA-patient cells (Figure 6b,d). We hypothesized that this phenotype may be caused by a CSA 8M folding/stability issue and therefore examined its interaction with TRiC chaperonin. Indeed, we found that the CSA 8M mutant more strongly associated with the TRiC complex, indicative of a protein folding/stability issue.

3. Page 11, the authors conclude “that TRiC, by regulating CSA stability and incorporation into the CRLCSA complex, promotes TC-NER and protects cells against UV-induced damage”. The reduced viability after UV irradiation of cells depleted for TCP1 (Figure 6C) could be due to another substrate of TRiC. Since incorporation of CSA into the CRL complex is only mildly reduced by depletion of TRiC, the authors should validate their model by testing if mild overexpression of CSA will complement the UV sensitivity caused by depletion of TCP1.

We have performed the requested experiment by overexpressing CSA-FLAG in TCP1-depleted Tert-immortalized primary human fibroblasts (VH10-Tert). Indeed, overexpression of CSA-FLAG partially rescues the UV-sensitivity of TCP1-depleted cells (see new Supplementary Figure 6b), further corroborating that increased UV-sensitivity by TCP1-depletion is (at least in part) explained by the reduction of functional CSA.

Minor question:

1. Do the FLAG- and GFP-tagged versions of CSA complement?

FLAG-tagged CSA complements the TC-NER and UV-survival defects of CSA-deficient patient cells (Figure 6b and 6d). We added new results showing that FLAG-tagged CSA complements the UV-survival defect of CSA-deficient patient cells to that of wildtype VH10-SV cells (Supplementary Figure 6b). Moreover, we added new results showing that GFP-tagged CSA also fully rescues the survival defect of CSA knockout U2OS cells following exposure to Illudin S, which induces transcription-blocking DNA lesions that are repaired by TC-NER (Figure 4c and 6e). Thus, FLAG and GFP-tagged CSA fusions complement and are fully functional, which is mentioned now in the text.

Reviewers' Comments:

Reviewer #1 (Remarks to the Author):

The authors have satisfactorily addressed most comments and concerns raised by the reviewers.

Reviewer #2 (Remarks to the Author):

Review of manuscript NCOMMS-17-19665A

The revised manuscript entitled “TRiC controls transcription resumption after UV damage by regulating Cockayne Syndrome protein A” by Pines and colleagues fully addresses all my concerns regarding the original manuscript. This is an excellent body of work describing the importance of proper assembly of DNA repair complexes exemplified by the assembly of CSA into the CRL complex by TRiC.